# Autonomous Visual Perception for Unmanned Surface Vehicle Navigation in an Unknown Environment

**DOI:** 10.3390/s19102216

**Published:** 2019-05-14

**Authors:** Wenqiang Zhan, Changshi Xiao, Yuanqiao Wen, Chunhui Zhou, Haiwen Yuan, Supu Xiu, Yimeng Zhang, Xiong Zou, Xin Liu, Qiliang Li

**Affiliations:** 1School of Navigation, Wuhan University of Technology, Wuhan 430063, China; yqwen@whut.edu.cn (Y.W.); church_zhou@whut.edu.cn (C.Z.); hw_yuan@whut.edu.cn (H.Y.); sp_xiu@whut.edu.cn (S.X.); yimengzhang94@gmail.com (Y.Z.); zx2000@whut.edu.cn (X.Z.); 2Department of Electrical and Computer Engineering, George Mason University, Fairfax, VA 22030, USA; 3Hubei Key Laboratory of Inland Shipping Technology, Wuhan University of Technology, Wuhan 430063, China; 4National Engineering Research Center for Water Transport Safety, Wuhan University of Technology, Wuhan 430063, China; 5Institute of Ocean Information Technology, Shandong Jiaotong University, Weihai 250357, China; axinzaixian@163.com; 6School of Electrical and Information Engineering, Wuhan Institute of Technology, Wuhan 430205, China; 7School of Transportation, Wuhan University of Technology, Wuhan 430063, China

**Keywords:** unmanned surface vehicles, vision, recognition, deep-learning, water region

## Abstract

Robust detection and recognition of water surfaces are critical for autonomous navigation of unmanned surface vehicles (USVs), since any none-water region is likely an obstacle posing a potential danger to the sailing vehicle. A novel water region visual detection method is proposed in this paper. First, the input image pixels are clustered into different regions and each pixel is assigned a label tag and a confidence value by adaptive multistage segmentation algorithm. Then the resulting label map and associated confidence map are fed into a convolutional neural network (CNN) as training samples to train the network online. Finally, the online trained CNN is used to segment the input image again but with greater precision and stronger robustness. Compared with other deep-learning image segmentation algorithms, the proposed method has two advantages. Firstly, it dispenses with the need of manual labeling training samples which is a costly and painful task. Secondly, it allows real-time online training for CNN, making the network adaptive to the navigational environment. Another contribution of this work relates to the training process of neuro network. An effective network training method is designed to learn from the imperfect training data. We present the experiments in the lake with a various scene and demonstrate that our proposed method could be applied to recognize the water region in the unknown navigation environment automatically.

## 1. Introduction

The past decade of research in marine and field robotics gave rise to the development of a new class of small-sized USVs. Currently, the USV is playing more and more important roles in maritime applications, such as meteorological monitoring, maritime search and rescue, geophysical exploration, environmental monitoring [1,2]. As a vision sensor, the video cameras are equipped in almost all of USVs to accomplish a variety of missions [3,4,5,6]. Although by itself camera is data-rich, how to extract useful information from millions or billions of raw pixels remains to be a challenging problem. One of those specific to USVs is water region segmentation, i.e., assigning each pixels a meaningful label such as water, sky, shore, or other vehicles in view, even a coarse water-none-water labeling is very helpful to the auto-navigation task of USV.

The water region recognition in the image is of great importance in many applications, such as obstacle detection and avoidance, path planning [7]. The target search could be constrained in the specific region by the water region segmentation to reduce the computation. In the past, a lot of researchers directly focus on the detection of the water-shore or water-sky boundary by analyzing low level image features such as color, edge or texture [8,9,10,11]. In [11], the Otsu’s thresholding algorithm was applied to threshold the images into binary images. Then, the sea-sky line was extracted with Hough transform as the longest straight line. Zafarifar et al. [12] used a method based on edge and color to detect the horizon line in an image. In [13], a hierarchical horizon detection algorithm was proposed by using Canny edge detector and Hough transform to find major edges in the image, and the precise curvature of the horizon was captured by fine-level adjustment. Lu et al [14] designed a modified edge detection algorithm according to the traditional Canny algorithm. Least square optimization is also a staple regression method for line detection but this method is more sensitive to noise than the Hough transform based methods [15]. Wang et al. [16] sampled maximum or minimum gradient value points along each vertical line in the gradient image, and then used RANSAC to fit the horizon line.

With the invention of novel algorithms and great improvement of computing power, deep learning based algorithms has achieved remarkable performance in many computer vision tasks, such as visual recognition [17], video processing [18,19] and semantic segmentation [20]. It has already been applied to the USV in the navigation, obstacle avoidance and environment perception [21,22]. Semantic segmentation as one of the widest fields of the application improves the scene recognition capability of the USV. Instead of only focusing on the contour boundary, these methods classify all pixels of the image. Fefilatyev et al. [23] proposed to segment the sky and then detect the horizon line in the binary mask of the network output. Cheng proposed the SeNet [24] to segment sea-land region, which combined the segmentation task and edge detection task into an end-to-end deconvNet in a multi-task way. However, the methods based on the state-of-the-art deep learning heavily rely on the manual label training data set.

In this paper, we propose a novel online learning approach to recognize the water region for the USV in the unknown navigation environment. Our method is divided into two main steps. Firstly, the water surface region is predicted by the proposed multi-stage adaptive segmentation method. Secondly, the predicted map is used as labeled training samples to train our segmentation network. Then the online trained network is used to segment the input image with a post-processing of conditional random field (CRF) to refine and improve the result. The rest of the paper is organized as follows: Section 2 describes the proposed algorithm in detail; Experimental dataset and results are demonstrated and compared in Section 3; Section 4 concludes this work.

## 2. Proposed Method

In this section, the proposed method for the dynamic water surface segmentation is explained in detail. The diagram of the proposed algorithm is shown in Figure 1. The free safe water region is detected by an obstacle avoidance sensor. In our experiment, a lidar sensor is used to perceive the obstacle in front of the USV. Based on the spatial relationship between the lidar sensor and camera, we can determine whether there are obstacles in the special region of the image corresponding to the sensing range of the lidar. Then, each pixel in the special region of the image is confirmed whether they are obstacles or just water surfaces.

Meanwhile, the adaptive segmentation method is used to segment the image into different regions based on the similarity in color and texture. Different segmentation results are obtained by adjusting the sensitivity of two region merging. By adjusting the sensitivity of the region merging, we get multiple segmentation of the image. Then, the self-generated label and the corresponding weight map are generated according to the multi-stage segmentation and the free safe water region. Finally, the binary cross entropy loss function, which takes into consideration the diverse uncertainty in different regions, is designed to train the network. When the segmentation network is used for prediction, the conditional random field (CRF) is used to refine the network output.

### 2.1. Adaptive Segmentation

In this article, a modified and enhanced version of graph-based image segmentation technique adapted from algorithm proposed by Felzenszwalb [25] is used to segment the input image initially. Compared with the k-mean based segmentation method that takes the global color statistical characteristic into consideration, the graph-based method is based on the local color and texture feature. In a graph-based segmentation approach, each pixel is regarded as a separate region initially, then each of those small regions are merged into super region recursively until specified ending criteria is satisfied. The following is a detailed introduction of the region merging process. 

G=(V, E) is an undirected graph with vertices $v_{i}\in V$, the set of image pixels to be segmented, and edges $e\left(v_{i},\;v_{i}\right)\in E$ corresponding to pairs of neighboring pixels. Each edge $e\left(v_{i},\;v_{i}\right)$ has a corresponding weight $w\left(v_{i},\;v_{j}\right)$, which is a non-negative measure of the dissimilarity between the two pixels connected by that edge (e.g., the difference in color, location or some other local attribute). A segmentation S is a partition of *V* into components such that each region C∈S corresponds to a connected region in a graph $G^{\prime }=\left(V,E^{\prime }\right)$, where $E^{\prime }\subseteq E$. Ideally, edges between two pixels in the same region should have relatively low weight while edges between pixels in different region should have higher weight. 

A predicate, *D*, is defined for measuring the evidence for a boundary between two regions in a segmentation. The predicate is based on comparing the dissimilarity between regions along the boundary of the two regions relative to the dissimilarity among neighboring regions within each of the two regions. The dissimilarity within each region C∈V is defined to be the largest edge weight of the minimum spanning tree in the region, MST(C, E):(1)$$\mathrm{Int}\left(C\right)=\underset{e\in MST\left(C,E\right)}{\max}w\left(e\right)$$

The external dissimilarity is defined by the difference between the two regions $C_{1},\;C_{2}\subseteq V$ to be the minimum edge weight connecting the two regions:(2)$$\mathrm{Dif}\left(C_{1},\;C_{2}\right)=\underset{v_{i}\in C_{1},v_{j}\in C_{2},\left(v_{i},v_{j}\right)\in E}{\min}w\left(\left(v_{i},\;v_{j}\right)\right)$$

Whether there is a boundary between a pair region is determined by checking if the difference between the regions, $\mathrm{Dif}\left(C_{1},\;C_{2}\right)$, is larger compared with the internal difference within at least one of the regions, $\mathrm{Int}\left(C_{1}\right)$ and $\mathrm{Int}\left(C_{2}\right)$. The pairwise comparison is defined as:(3)$$D\left(C_{1},C_{2}\right)=\{\begin{array}{cc} true & if\;\mathrm{Dif}\left(C_{1},\;C_{2}\right)>\mathrm{MInt}\left(C_{1},C_{2}\right)\\ false & otherwise \end{array}$$
where Mint(·) is defined as:(4)$$\mathrm{MInt}\left(C_{1},C_{2}\right)=\min(Int\left(C_{1}\right)+\tau \left(C_{1}\right),Int\left(C_{2}\right)+\tau \left(C_{2}\right)$$

The function τ controls the degree to ensure that the difference between the two regions must be greater than their internal differences. To avoid the influence of the small region, the function τ is defined with a factor of the region size:(5)$$\tau \left(C\right)=\frac{k}{\left|C\right|}$$
where |C| denotes the size of C, and the parameter k is a constant.

The size of the segmented region is increased as *k* grows. Due to the various light intensity and visibility of the water environment, the algorithm has a poor performance with the constant parameter *k*. To make the algorithm robust to the different condition, the τ(C) is modified as:(6)$$\tau \left(C\right)=\frac{k_{0}\cdot \sigma_{I}}{\left|C\right|}$$
where $\sigma_{I}$ is the mean standard deviation of the image pixel value in each channel:(7)$$\sigma_{I}=\frac{1}{3}\sum_{i=r,g,b}\sigma_{i}$$
(8)$$\sigma_{i}=\sqrt{\frac{\sum \sum {\left(p_{i}-{\bar{P}}_{i}\right)}^{2}}{H\cdot W}}$$
where *H* and *W* are the height and width of the image. By relating the controlling parameter *k* to the standard deviation of the image pixels, the modified algorithm is able to achieve better segmentation performance in adversary situation such as scene with low contrast or low visibility

### 2.2. Water Region Prediction Based on Multi Stage Segmentation

In order to classify the segmented region into water or none-water surface, a lidar device is used to sense a small region within close range of USV; the corresponding patch in the image is assumed to be water surface when there is no reflection signal from lidar. The segmented region of similar color and texture as this special region are then merged together and labeled as water surface; all the other segmented regions are labeled as none-water region:(9)$$c\left(x,y\right)=\{\begin{array}{cc} 1 & if\;\forall p\left(x_{m},y_{m}\right)\in (s_{i}\cap s_{R}),\;c\left(x_{m},y_{m}\right)=1\\ 0 & \mathrm{otherwise} \end{array}$$

The equation above describes the process of the water region prediction. $s_{i}$ is a region of the image segmentation *S* and $p\left(x,\;y\right)\in s_{i}$. $s_{R}$ is the region that is corresponding to the sensing range of the lidar. If there is a point belonging to $s_{R}$ and confirmed as the water region, then any point in the region is regarded as the water surface. Otherwise, it’s regarded as the non-water-surface.

During the adaptive segmentation, the tendency of two regions merging into one region decreases as the value of the function *τ* increases. When *τ* is small, the segmentation is of fine grain and segmented regions are of small size. The prediction whether a region connected to the convince water surface is also the water region have high confidence. And there is a high possibility that many of the regions regarded as the non-water-region are indeed the water, i.e., a miss. On the other hand, when *τ* is large, the segmentation is coarse and the segmented regions are of large size. There is a high possibility that parts of the large regions connected to the convince water surface are non-water, i.e., a false alarm. To reduce the dependency on empirically adjusting parameters, multiple value of *τ* are used and combined to generate a probability map:(10)$$\tau_{n}\left(C\right)=f_{n}\frac{k_{0}\cdot \sigma_{I}}{\left|C\right|}$$
(11)$$f_{n}=1.5^{n-1}$$
where $f_{n}$ is a factor that controls the sensibility of the segmentation and n∈{0, 1,⋯, 9}. Through the multistage segmentation with the different value of $f_{n}$, each pixel of the image could have a set of labels as define by Equation (9), $c\left(x,y\right)=\left\{c_{0},\;c_{1},\cdots ,\;c_{10}\right\}$. Then the probability of the pixels that belongs to the water surface is defined as:(12)$$q\left(x,y\right)=\frac{1}{10}\sum_{i=0}^{10}c_{i}$$

### 2.3. Segmentation Network

The label maps generated previously by multistage segmentation are used as training samples to train the segmentation network that is based on U-Net [26] which is a very popular end-to-end encoder-decoder network for semantic segmentation. The U-Net structure uses multiple feature maps in different layers and has a good performance in different applications.

Each layer in the encoder network performs convolution with a filter bank to produce a set of feature maps. The max-pooling with a 2×2 window and stride 2 is performed to achieve translation invariance over small spatial shifts in the input image. The output is sub-sampled by a factor of 2 in each max-pooling operation. This results in a large input image context for each pixel in the feature map. In order to use the available pre-trained model, the last fully connected layers of VGG16 [27] are eliminated and the fully convolutional layers of VGG16 are reserved as the encoder. Based on the feature map size, the convolution layers are divided into six stages. The detailed structure of the designed network is shown in Figure 2.

### 2.4. Online Training

A cross entropy loss function is used to train the network. *N* is defined as the number of image pixels used to update the network parameters. y(x,y) and $\hat{y}\left(x,y\right)$ are defined as the labels of the pixel p(x,y) predicted by the proposed multi-stage adaptive segmentation method and the corresponding predicted probability from the network respectively. The binary cross entropy loss is then expressed as:(13)$$L=-\frac{1}{N}\sum^{N}\left[y\left(x,y\right)log\hat{y}\left(x,y\right)+\left(1-y\left(x,y\right)\right)log\left(1-\hat{y}\left(x,y\right)\right)\right]$$

As the label is auto-generated with uncertainty, the training with any wrongly labeled data results in network degradation. To improve the training efficiency, we emphasize on the pixels with high confidence during the training.

The value calculated by Equation (12) is interpreted as the possibility of being classified as water region, where value near 1 means confirmation of water region and 0 means confirmation of none-water region and any value near 0.5 rendering the prediction is unreliable. Based on the pixel prediction, a confidence map is generated to represent the reliability of each pixel prediction. The confidence map is estimated as:(14)$$w\left(x,y\right)=\frac{cos\left(2\pi \cdot q\left(x,y\right)\right)+1}{2}$$

A cosine function is the mapping function. As q(x,y) varies from 0 to 1, w(x,y) has a high value at both ends and a low value in the middle. Thus, the confidence map is regarded as the weight map to address the problem of the reliability imbalance in different pixels. The binary cross entropy loss is modified as:(15)$$L=-\frac{1}{N}\sum^{N}w\left(x,y\right)\left[y\left(x,y\right)log\hat{y}\left(x,y\right)+\left(1-y\left(x,y\right)\right)log\left(1-\hat{y}\left(x,y\right)\right)\right]$$

The advantage of this loss function is that it puts more weight on pixels with greater label confidence and less weight on those unreliable pixels, hence reduces the degradation of network caused by bad training samples.

### 2.5. Segmentation Improvement

In the field of image multi-class object segmentation, the conditional random field (CRF) is an effective way to classify the pixel category by considering the relationship between a single pixel and all other pixels in an image [28]. In this paper, a method adopted from the work of Deeplab [29] that combined CNN and CRF, is used to further improve the output possibility map of the segmentation network. The energy function of the CRF model is:(16)$$E\left(x\right)=\sum_{i}\varphi_{i}\left(x_{i}|y\right)+\sum_{i,j}\phi_{i,j}(x_{i},x_{j}|y)$$

As in the standard CRF-based image semantic segmentation [30], each pixel is represented as a random variable. Each of these random variables takes a label from the set $L=\left\{l_{1},l_{2}\right\}$, which represent the water region and the others. Let $X=\left\{X_{1},X_{2},\cdots ,X_{N}\right\}$ denote the set of random variables corresponding to the image pixels. *N* is equal to H×W (H is the Height and W is the width of the image). A labeling *x* refers to any possible assignment of labels to the random variables and takes values from the set *L*. In the given image, y represents the image pixel value (or the features map extracted from the small area around pixel *i*), and $x_{i}$ represents the random label of pixel *i*.

#### 2.5.1. Unary Potentials

(17)$$\varphi_{i}\left(x_{i}|y\right)=-logP(x_{i}|y)$$

The unary potential is defined as the negative log of the likelihood of a label being assigned to pixel *i*. It can be treated as local classifiers and well defined by the output of the network model, a probability map for each class in each pixel.

#### 2.5.2. Pairwise Potentials

The pairwise potential models the relationship among neighboring pixels and weighted by color similarity. The pairwise potential is defined as the following:(18)$$\phi_{i,j}\left(x_{i},x_{j}|y\right)=u\left(x_{i},x_{j}\right)\left[\omega_{1}exp\left(-\frac{{∥p_{i}-p_{j}∥}^{2}}{2\sigma_{\alpha }^{2}}-\frac{{∥y_{i}-y_{j}∥}^{2}}{2\sigma_{\beta }^{2}}\right)+\omega_{2}exp\left(-\frac{{∥p_{i}-p_{j}∥}^{2}}{2\sigma_{\gamma }^{2}}\right)\right]$$
where $p_{i}$ and $p_{j}$ are the pixel position. $u\left(x_{i},y_{j}\right)$ represents the compatibility of the labels assigned to variables $x_{i}$ and $x_{j}$, where $u\left(x_{i},x_{j}\right)=0$ if $x_{i}=x_{j}$ and 1 otherwise. The first term depends on both pixel positions and pixel interactions with the neighboring information. And the second term only depends on pixel positions.

Each random value of the CRF is initialized as the corresponding output from the CNN segmentation network described in Section 2.3. Then the optimization result is obtained by minimizing the above CRF energy E(x). CRF refines the local segmentation and improve the algorithm robustness by considering the color and spatial information.

## 3. Experimental Results

To verify the performance of the proposed method, several experiments and comparison were carried out based on the data captured by an experimental USV navigating in East Lake located in Wuhan, China. Figure 3 shows the experimental USV and the navigation scene. Table 1 shows the detail of the experimental USV and computer.

### 3.1. Adaptive Segmentation

The proposed adaptive segmentation is based on the graph-based segmentation algorithm. To demonstrate the robustness of our multistage adaptive segmentation algorithm, we compare our method with the other similar graph-based segmentation algorithm [25] which has a standard implementation in the popular OpenCV library. As shown in Formula (5), there is a parameter k controlling the grain level of the segmentation, so we run the algorithm multiple times with different k value and compared the results with that of the proposed method.

Four typical lake scenes were used as the testing case. For each scene, the minimum region size of segmentation was set as 100, but three different k value were tried, k = 600, k = 1200 and k = 2000. The result of different algorithms or parameters are tiled in Figure 4. As shown in the first and second rows of the figure, the segmentation is too coarse with a large k for low contrast scene. As shown in the third and fourth rows of the figure, the segmentation is too fine with a small k for normal contrast scene. However, our proposed method could segment all of these images properly without tuning any parameter, neither too fine nor too coarse in any case.

### 3.2. Process of the Training Data Generating

This experiment shows the process of the water region prediction. Based on the proposed adaptive segmentation algorithm, ten segmentations from fine to coarse are obtained. Based on the lidar information and the multistage segmentation result, the water region prediction map is generated by fusing maps from different stage as Equation (12), then the weight map is generated as Equation (14). Figure 5 shows the final prediction map and the corresponding weight map. The prediction maps and the weight maps are shown by the second and third row of the images, respectively.

### 3.3. Performance of the Proposed Method

Our goal is to make USV automatically navigate in the unknown environment and learn the water region during exploring. Figure 6 shows the performance of our online method during navigation. In the beginning, our segmentation network does not have any prior-knowledge of the water region. It cannot properly recognize the water surface. The segmentation performance is continuously improved as online training progresses, as shown in the first row of Figure 6. After sailing for a while, our system is adapted to the unknown environment and could recognize the water surface region. The second row of Figure 6 shows the segmentation results of images in different perspectives after enough training.

The performance of this approach can be better understood by the comparison with the state-of-the-art methods for semantic video segmentation. The first set of baseline methods, K-mean and graph-based segmentation method, are two representative conventional methods, which segment the image according to the feature of the pixel value and region edge. The second set of methods, UNet [26], RefineNet [31] and DeepLab [32], are Convolutional Neural Networks that need training with large labeled data.

Quantitative results are provided in Table 2. The segmentation quality is measured using average segmentation precision. In each frame, we measure the intersection divided by the union of the segmentation result and the ground truth labeled manually. Then, we average those scores over all frames in the sequence. Five videos of about 20–30 min in length recorded at different times and from different scenes are used to compare the performance of these algorithms. Each video is converted to images and about ten thousand images are selected as the experiment data. Twenty percent of the images are labeled as the test data in each video. For the traditional method, an image is segmented into different region according to the pixel value for the k-mean method and the image edge information for the graph-based method. The region connected to the free region in front of the USV is regarded as the water surface region. Segmentation precision is measured according to the corresponding test label. For the methods based on deep learning, eighty percent of the images are labeled as the training data in each video. Then these convolutional Neural Networks are trained with the training label data and tested by the test data. With our method, no pre-labeled training data are needed and learning is automatically. After the training is completed by itself, the test data is used to test our method.

As the traditional method is susceptible to the factors, such as waves, reflections, and illumination, the K-mean and graph-based methods have a poor performance. With enough labeled training data, the UNet, RefineNet and DeepLab perform well with high precision. However, the performances of these methods heavily depend on the training data set. Due to the complexity and variability of the navigation environment, only limited scenes are sampled and labeled as the training data. Without any pre-labeled training data, the performance of the proposed method is close to the state-of-the-art. With the continuous online learning, the proposed method is robust in the rich and varied scene. This can be manifested in testing case video3 and video5 where the proposed method shows better performance compared with UNet.

## 4. Conclusions

In this paper, an online semi-supervised learning method for water region segmentation is proposed for USVs in a dynamic navigation environment. Automatically labeled training data are generated by an adaptive multistage segmentation method with the aid of lidar data. An encoder-decoder semantic segmentation network is designed as the main work horse for the task. A modified binary cross entropy loss function with a weight map was applied to address the reliability issue in different pixels during the network training. Our experimental results showed that the proposed method outperformed the conventional methods. While having a similar performance as the state-of-art deep learning based algorithms, the proposed method has the advantage of self-training without the need of manually labeled data. With incremental and continuous learning online, the proposed method is robust and adaptive to the changing unknown environment.

In this paper, we verified the validity of the proposed method. However, we did not address some very hard situation for vision navigation, for example, water surface mirror reflection near shore. In some extreme cases, even a human eye can barely tell the difference between the shore objects and its mirror reflections on the water surface. Future work may direct to attack those difficult problems by multi-sensor data fusion and time-sequence deep learning algorithm such as recursive network (RNN).

## Figures and Tables

**Figure 1 sensors-19-02216-f001:**
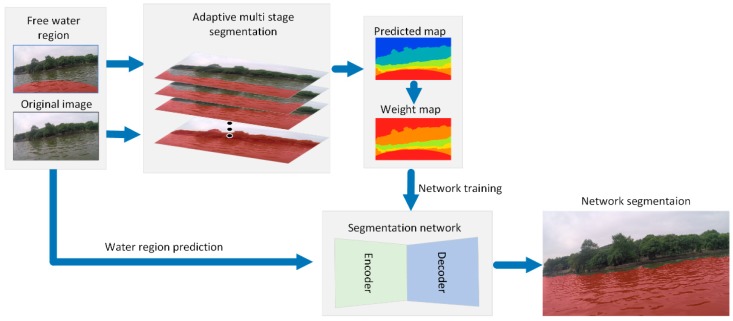
The diagram of the proposed method.

**Figure 2 sensors-19-02216-f002:**
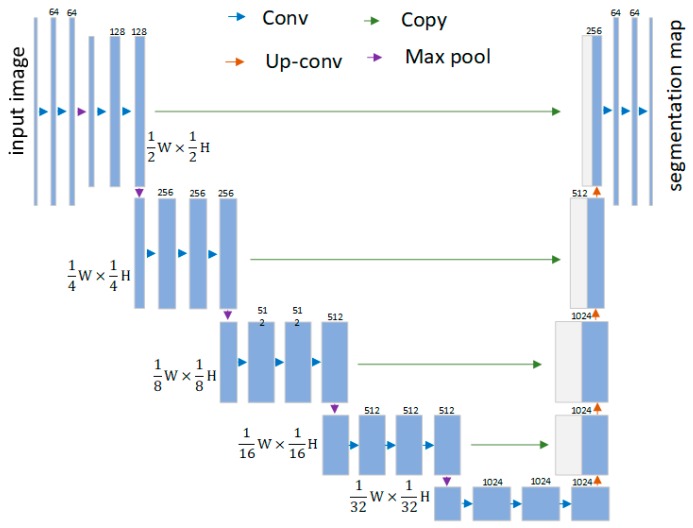
Segmentation network.

**Figure 3 sensors-19-02216-f003:**
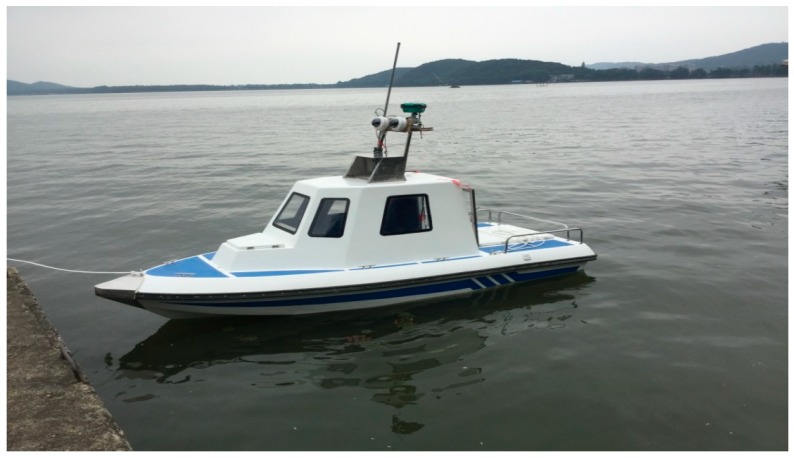
The experimental USV.

**Figure 4 sensors-19-02216-f004:**
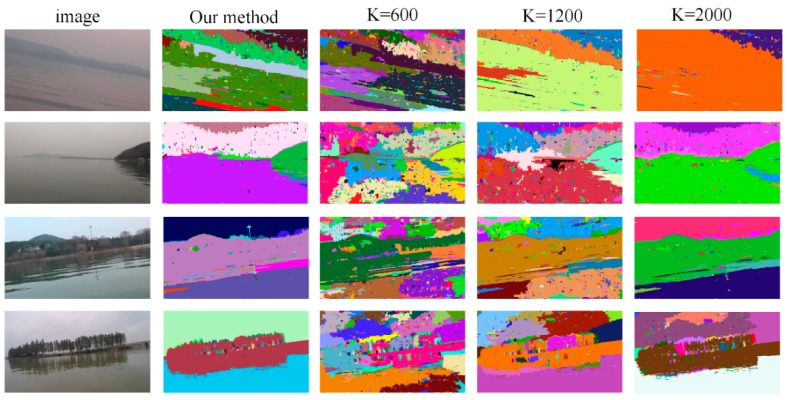
Adaptive segmentation result comparision.

**Figure 5 sensors-19-02216-f005:**
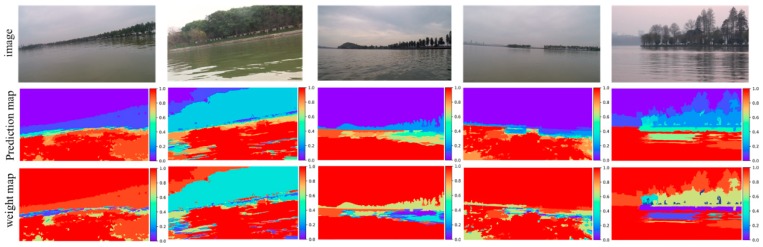
Water region prediction and weight map generating.

**Figure 6 sensors-19-02216-f006:**
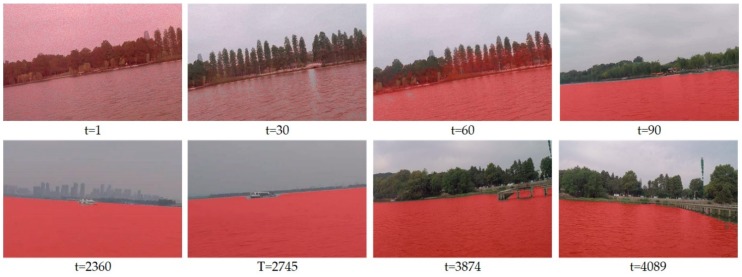
Segmentation performance during the online learning.

**Table 1 sensors-19-02216-t001:** Parameter of the experimental USV and computer.

Equipment	Parameter	Value/Model
USV	length	3.96 m
	width	1.55 m
	draft	0.3~0.5 m
	Max speed	2.2 m/s
Computer	CPU	Intel i7-5820
	GPU	Titan X
	Memory	32 GB

**Table 2 sensors-19-02216-t002:** Precision of the state-of-the-art methods and the proposed method.

Method	Video1	Video2	Video3	Video4	Video5
K-mean ^1^	60.1	65.2	59.5	62.7	67.2
Graph-based ^1^	73.6	64.5	56.7	63.5	71.4
UNet ^2^	97.9	95.3	96.2	97.1	95.8
RefineNet ^2^	9.5	97.6	98.1	97.9	98.4
DeepLab ^2^	99.7	9.5	78.7	99.2	98.7
Ours ^3^	97.3	94.3	97.2	96.4	96.3

^1^ The traditional feature-based method; ^2^ Methods based on deep learning with manual label training data set; ^3^ The proposed method that is self-learning with no manual label data set.

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
