# Peer review of "Autonomous Visual Perception for Unmanned Surface Vehicle Navigation in an Unknown Environment"

_sensors, 2019, doi:10.3390/s19102216_

Round 1
Reviewer 1 Report
The general comment to the paper is that it is very interesting and worth reading.
Remarks:
1. I am not a native speaker but I found in the paper at least several spelling errors, e.g. “Five videos recorded in different time and scene is used”.
2. In section 2.1 authors describe their method of adaptive segmentation. It gives a method of evaluating a segmentation if the image is already segmented, that is, if each pixel is already assigned to a region. How pixels are assigned to regions is not described.
3. In section 2.5, the application of CRF to improve image segmentation is outlined. For readers who are not familiar with this method it is very difficult to understand how improvement is performed. It seems that it would be great, for convenience of each reader, to add some details of CRF and its application in author’s method.
4. At the beginning of section 3.1 authors mention that they compare their adaptive segmentation to an OpenCV method, but which method is not given. Then, in Table 1 K-means is mentioned but this information should be given earlier.
5. In the same paragraph, parameter k is mentioned, but it is difficult to say, whether it is K-means parameter or adaptive segmentation parameter (eq. 5). From the context it appears that this is K-means parameter but it should be clearly stated.
Author Response
1.Thank you for your advice. I have checked the language and correct many mistakes. Helps and suggestions are greatly appreciated if more mistakes are found and pointed out.
2.Firstly, each pixel is regarded as a separate region. Then each of those small regions are merged into super regions recursively until the ending criteria is met. This description is added at the beginning of section 2.1 according to your suggestion.
3.CRF is described in detail to help readers understand the algorithm. And How to apply the method in our work is mentioned in the last part of section 2.5.
4. I have added the reference paper to the compared method. This agorithom is implemented in the opencv API of Graph Based Segmentation Algorithm is implemented in the opencv API class of cv::ximgproc::segmentation::GraphSegmentation Class. And the k-mean method is mentioned in the beginning of section 2.1 to compare with the graph-based method. K-mean method presents the method that uses the statistical color infomation. And the graph-based method takes advantage of both the color and the location information
5.Parameter k is described in formula 5 in paragraph 2.1. As your suggestion, I have added the reference formula to the parameter k at the beginning of section 3.1
Reviewer 2 Report
Introduction:
---------------
A few more papers need to be referred while saying about navigation part in the starting paragraph where the use of different path planning algorithms needs a mention.
Some recent review papers can actually make it good for readers.
Some important and recent studies in this regard are :
-Singh, Yogang, Sanjay Sharma, Daniel Hatton, and Robert Sutton. "Optimal path planning of unmanned surface vehicles." Indian Journal of Geo-Marine Sciences 47, no. 07 (2018): 1325-1334.
A mention of a few other navigation studies based on AI techniques in the introduction is required to give readers a brief overview of methods in AI.
Some mention is:
- Singh, Y., Sharma, S., Sutton, R., Hatton, D., & Khan, A. (2018).
A constrained A* approach towards optimal path planning for an unmanned surface vehicle in a maritime environment containing dynamic obstacles and ocean currents. Ocean Engineering, 169, 187-201.
Proposed Method:
--------------------
No comment has been made on computational effort towards online training on USV has been made.
Which USV is proposed and how does current methodology of CNN is effective in long endurance mission of USVs. There is a need to comment on that.
USVs are subjected rules of the marine system; where COLREGs implementation is imperative. How can we integrate COLREGs with such a method? Comment that.
What is the number of data set used to train the data set? There is no mention. If the dataset is less than 10,000; can we say that the trained network will work in all conditions?
Experimental Results:
----------------------
When the author says a different experiment; what he means by that and how he classifies different cases studies !!
Do different case studies have different weather and different lightning while taking test cases and using them for training !! Comment on that
Table 1 compares against different methods but is those methods adopted for USV navigation with the same dataset and if not then how this comparison of performance is valid !! comment on that
What happens when USV is taking pictures in harsh conditions. Comment on that and if the online training can take care of that !!
Recommendation: Accept with major revision
Author Response
Introduction:
Thank you for your good advice. I've included the suggested papers into the reference in the revised paper. Although my research is primarily focused on perception, those papers are inspiring and worth reading. I also refered them to my collegues who are doing more research on USV path planning.
Proposed Method:
1. At the present stage, we focused on verification of our methodology ideas and do not concern too much about the problem of computational load. All data were captured during field experiment and processed offline later by a desktop-computer with Titan X GPU. Each video is of resolution 1920x1080 and is about 20-30 minutes. Each video is converted to ten thousand images to train the network. The data processing took about several days including training. Computation could be reduced in two ways by a quick fix: 1, reduce the image resolution, since in most navigation scenario high resolution may not be needed. 2, downsampling in time, since most USV navigation scenario will not involve drastic maneuver. Other techniques to reduce the computation load is also possible, such as reducing the network complexity. We'll tackle those problem in future work.
2.Our designed USV is used to capture the data. Our method is designed for the life-long learning. The capability of online learning is actually shown in figure 5. The CNN network was initialized randomly without any prior knowledge. As the USV navigated through different part of the lake, scene view and illumination changed constantly, training samples were automatically generated and fed into the segmentation network for training, gradually the segmentation network learn the feature of water surface and was able to segment the water surface.
3.That is a good comment. In this paper, we focus on the task of USV visual perception and scene unstandering. Our experiments were conducted in a lake where COLREGs is not a concern or simply does not apply. Of course, COLREG is a must for any practical application in most water way. To incorperate COLREG, additional information and sensor data need to be fused. First, AIS/Radar data need to be fused smoothly with vision data to identify and track target and measure the distance and bearing angle, visual object recognition and tracking can be combined with radar target tracking and AIS target identification. This is a very interesting and challenging research topic and there is another Phd student in our team is working on this currently. Secondly, IMU data need to be exploited to build the geometrical projection model, this project model will be used to build a dynamic local navigational map to aid the decision making process for collision avoidance or local path planning or ship motion maneuver. Currently we are building a larger USV capable sailing in sea water and we're certainly considering integrating the COLREGs implementation.
4.I have added the detail of the data set in the third paragraph in section 3.3 according to your question. In this paper, the videos are captured from several navigation experiment in the lake. Five videos are selected to test our method. Each video is converted to ten thousand images first and then processed by the proposed method. By its nature, the proposed method is a kind of incremental online training, so presumbly , as long as the USV navigates long enough, it should be able to collect plenty of training data to cook the network well. However, it does FAIL in some extreme cases, one situation is water surface mirror reflection near shore, sometimes even a human eye can barely tell the difference between the shore objects and its mirror image on the smooth,quiet water surface of a inland lake. At sea, since most sea water surface are full of waves or bubbles, mirror reflection may not outstand itself much and is not a major problem. Sun glaring or night light glaring may pose another extreme difficult situation. Those are very interesting research topics and worth exploring. We can not cover all problems in one paper, more research will be conducted to address those mentioned issues and more.
Experimental Results:
1.The different case studies include different weather in different time except the raining day. The lightning changes as a result of the USV turning and the relative position relation to the offshore.
2. Thank your for your good question. I have found that I did not express it clearly in the third paragraph in section 3.3. These methods were tested with the same dataset but by different methods for comparison. Twenty percent of the images from each video are labeled as the test data to test all the methods. The traditional methods are directly tested with the testing data.
Eighty percent of the images from each video are labeled as the training data. Then the compared deep learning methods were training with the training data. After the training, the networks are test with the test data. In the proposed method, the images are directly used as the training data. After the self-training, the propose method is tested with test data set.
3. That is a good question. For now, we just focus on proof of concept and verifying of our basic ideas. We have found that the error comes mainly from the harsh conditions, especially the ashore water reflection. This is a particular problem for sailing in the inland water close to shore. In the process of the prediction, the water reflection regions have low confidence. Our segmentation network was trained with low weight in these regions. That is to say, these regions do not have enough training, which resulted a bad performance to these cases. Our next step research is focus on the improvement of prediction ability on harsh condition increasing the training on the complex areas. One idea would be exploiting the temporal property of dynamic texture of water surface. This mirror reflection problem is not serious when sail in sea water.
According to your question, I have added a few comments in the last paragraph of the conclusion section.
Round 2
Reviewer 2 Report
The authors have respected the comments and the manuscript looks good to be published